# SEE NO EVIL: Adversarial Attacks against Context-dependent Visual Association in Referring Multi-Object Tracking Systems

**Halima Bouzidi**[1]*, **Haoyu Liu**[1], **Mohammad Abdullah Al Faruque**[1]
[1]University of California, Irvine

## Abstract

Language–vision understanding has driven the development of advanced perception systems, most notably the emerging paradigm of Referring Multi-Object Tracking (RMOT). By leveraging natural-language queries, RMOT systems can selectively track objects that satisfy a given semantic description, guided through Transformer-based spatial–temporal reasoning modules. End-to-End (E2E) RMOT models further unify feature extraction, temporal memory, and spatial reasoning within a Transformer backbone, enabling long-range spatial–temporal modeling over fused textual–visual representations. Despite these advances, the reliability and robustness of RMOT remain underexplored. In this paper, we examine the security implications of RMOT systems from a design-logic perspective, identifying adversarial vulnerabilities that compromise both the linguistic-visual referring and track-object matching components. Additionally, we uncover a novel vulnerability in advanced RMOT models employing FIFO-based memory, whereby targeted and consistent attacks on their spatial–temporal reasoning introduce errors that persist within the history buffer over multiple subsequent frames. We present VEIL, a novel adversarial framework designed to disrupt the unified referring–matching mechanisms of RMOT models. We show that carefully crafted digital and physical perturbations can corrupt the tracking logic reliability, inducing track ID switches and terminations. We conduct comprehensive evaluations using the Refer-KITTI dataset to validate the effectiveness of VEIL and demonstrate the urgent need for security-aware RMOT designs for critical large-scale applications.

## 1 Introduction

Referring Multi-Object Tracking (RMOT) has recently emerged as a key advancement in intelligent perception, enabling systems to track objects of interest based on natural language descriptions Botach et al. [2022], Wu et al. [2023], Zhang et al. [2024a], Du et al. [2024], Nguyen et al. [2023], Chen et al. [2025], Chamiti et al. [2025], Kong et al. [2025]. This capability is particularly valuable for real-world applications such as robotic vehicles and surveillance systems, where collaborative human–machine interaction depends on flexible and intuitive queries. In practice, RMOT is already deployed at scale in defense infrastructures to track subjects of interest based on non-biometric descriptors such as body size, hair color, accessories, and clothing style, without requiring any facial identity O'Donnell [2025]. The core challenge of RMOT lies in resolving ambiguities that arise in both visual and linguistic domains Radford et al. [2021]. Visually, systems must handle occlusions, extreme viewpoint changes, and the presence of visually similar objects. Linguistically, they must interpret context-dependent or imprecise expressions Yu et al. [2016]. Ultimately, the task is to learn a robust multimodal representation that maps noisy visual and textual inputs into a shared latent

---

*Correspondence to <hbouzidi@uci.edu>

NeurIPS 2025 Workshop: 1st Reliable ML from Unreliable Data Workshop (Reliable ML 2025).

space, allowing semantic alignment between object features and natural-language descriptions, and thereby supporting accurate and unambiguous target identification and tracking Wu et al. [2022].

Modern End-to-End (E2E) architectures Botach et al. [2022], Wu et al. [2023] employ multimodal Transformers that use learnable object queries as dynamic placeholders for targets. These models operate through two deeply interwoven mechanisms. *First*, for spatio-temporal reasoning, object queries from the current frame $F_t$ perform cross-attention on the visual encoder's output features to update their appearance and location. They also perform self-attention with the set of object queries from frame $F_{t-1}$, allowing the model to propagate identity and model object dynamics implicitly, replacing classical state estimation (e.g., Kalman filter Sahbani and Adiprawita [2016]). This process mainly builds a temporal memory of object trajectories. *Second*, for language-vision fusion, initial queries are often modulated by a global query embedding from a text encoder (e.g., BERT). During the decoding process, these evolving object queries repeatedly perform cross-attention with token-level language embeddings, forcing the model to continuously ground specific linguistic attributes (e.g., '*red moving cars*', '*pedestrians on the left side*') to corresponding visual objects.

However, this powerful integration of fusion, memory, and reasoning creates sophisticated adversarial vulnerabilities within the model's high-dimensional optimization landscape. Adversaries can exploit this by crafting perturbations in the input pixel space that induce large, controlled displacements of object feature representations on the learned manifold. Such perturbations can be designed to trigger temporal discontinuities and semantic misalignments by manipulating the visual input. This strategy is effective because directly altering the textual query is neither practical nor sufficiently stealthy, whereas subtle modifications to visual inputs can more effectively compromise the fusion process and corrupt the model's ability to maintain a consistent temporal memory, leading to tracking failures.

Previous adversarial attacks Jia et al. [2020], Wang et al. [2021], Zhou et al. [2023] are not fundamentally designed to exploit inherent vulnerabilities in RMOT. They primarily target discrete and separate components of traditional tracking-by-detection (TBD) pipelines, such as detection and association modules that are entirely replaced by learned mechanisms in modern E2E architectures. Critically, they fail to address the core set-based, bipartite matching paradigm that underpins Transformer-based trackers. These models Botach et al. [2022], Wu et al. [2023] rely on the Hungarian algorithm for optimal label assignment during training, a mechanism completely different from the heuristics targeted by prior works. Furthermore, their vision-only loss functions are incapable of manipulating the model's behavior within the joint multimodal embedding space, which is precisely where the final referring decision is computed based on language-vision feature similarity.

Addressing this gap, we introduce, VEIL, a novel adversarial framework that directly targets the core architectural principles of Transformer-based RMOT. We formulate the attack as a compound optimization problem, where the crafted perturbation is guided by specific adversarial loss functions that synergistically combines two objectives. *First*, we introduce a targeted referring expression loss that exploits the linguistic-visual association mechanism to force the model to assign high referring confidence to objects that are semantically plausible but contextually incorrect. *Second*, we introduce a spatial-temporal reasoning loss that systematically attacks the limited-capacity temporal memory by maximizing temporal inconsistency between consecutive frame embeddings. By jointly optimizing these objectives, VEIL demonstrates a sophisticated attack that exploits both the finite memory capacity and attention dependencies. Our comprehensive evaluation on benchmark RMOT models like TransRMOT and TempRMOT shows that by targeting these core cognitive mechanisms, our attack achieves significantly higher success rates across standard multi-object tracking metrics.

## 2 Related Work

**Referring Multi-object Tracking (RMOT).** Distinct from conventional MOT, RMOT is a multimodal task that involves tracking only the specific object instance designated by a natural language query. Early works often rely on disjoint pipelines, combining an off-the-shelf MOT with a separate visual grounding model Wu et al. [2022]. E2E multimodal Transformers, notably, MTTR Botach et al. [2022], TransRMOT Wu et al. [2023], TempRMOT Zhang et al. [2024a] enable a new standard by performing E2E joint spatio-temporal reasoning and language-vision fusion within a unified decoder. These architectures utilize language-conditioned object queries that are continuously refined by attending to both visual features and linguistic tokens. More recently, Refer-GPT and variants Chamiti et al. [2025], Kong et al. [2025], Nguyen et al. [2023] have started exploring the prospect

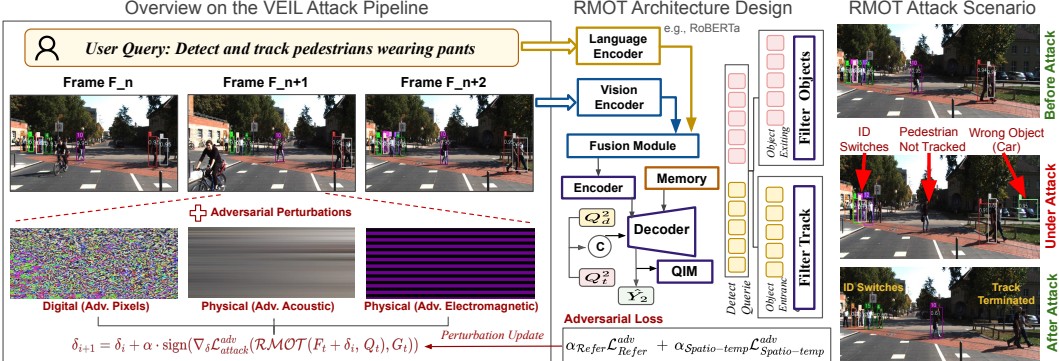

Figure 1: Overview of the proposed VEIL attack framework.

of leveraging large-scale Vision-Language-Model (VLM) to further improve the model's semantic understanding and tracking robustness in complex, open-world, and in-the-wild scenarios.

**Adversarial Attacks on Vision-Language Fusion.** Initial research on attacking tracking systems focused on vision-only (TBD models), targeting either the detector to induce false negatives/positives Wang et al. [2021], Zhou et al. [2023] or the association logic to cause identity switches Jia et al. [2020]. However, the model design assumptions of these attacks make them incompatible with modern E2E Transformer-based trackers grounded upon a different learning paradigm. More pertinent to RMOT is the related works on attacks against static VLMs. These attacks aim to break the learned alignment between modalities Long et al. [2022]. In Visual Question Answering (VQA), perturbations can force a model to fixate on irrelevant image regions and produce incorrect answers Yin et al. [2024]. For visual grounding, attacks have been shown to deceive models into localizing a completely different object from the one described in the text Wallace et al. [2019], Gao et al. [2024]. However, existing methods are designed for static, stateless tasks. The unique challenge of crafting a perturbation that can consistently deceive a dynamic, stateful RMOT across a video sequence, by specifically targeting its core bipartite referring-matching logic and temporal fusion mechanism, remains unexplored.

## 3  The VEIL Attack Framework

### 3.1  A Primer on Modern RMOT Architectures

Modern E2E RMOT architectures, like TransRMOT Wu et al. [2023] and TempRMOT Zhang et al. [2024a], are designed for joint spatio-temporal reasoning and language-vision fusion. These models consist of three components: a visual encoder, a text encoder, and a multimodal Transformer decoder that processes learnable object queries over time. The key innovation, particularly in TempRMOT, lies in how the decoder establishes a robust spatio-temporal memory for each tracked object.

**1. Feature Extraction.** Given a video stream and a referring language expression (from user query), the model first extracts high-level features from each modality.

**Visual Encoder:** For each video frame $I_t \in \mathbb{R}^{H_0 \times W_0 \times 3}$ at time $t$, a convolutional neural network (CNN) backbone, such as a ResNet-50, is used to extract a rich visual feature map. This map is then flattened and supplemented with a fixed positional encoding to retain spatial information, resulting in a sequence of visual features $F_v \in \mathbb{R}^{HW \times C}$, where $C$ is the feature dimension.

**Text Encoder:** The input language query, a sequence of words, is tokenized and fed into a pre-trained Transformer-based text encoder like BERT. This produces a sequence of contextual word embeddings $F_l \in \mathbb{R}^{L \times C}$, where $L$ is the length of the token sequence.

**2. Multimodal Transformer Decoder.** This is the core of the architecture, which takes a set of learnable object queries $Q \in \mathbb{R}^{N \times C}$ as input. The decoder consists of a stack of identical layers, each performing a sequence of attention operations to update the object queries $Q_t$ at the current frame $t$. A single decoder layer performs three key attention steps:

**Temporal Cross-Attention:** To propagate identity and motion information, a temporal fusion is performed. The queries from the current frame attend to a set of historical queries, effectively allowing the model to aggregate and refine a spatio-temporal memory of the object's past. While models

like TransRMOT only perform this with queries from the immediately preceding frame, advanced architectures such as TempRMOT use a dedicated module to create a more robust, long-term memory from a history of multiple past frames. This replaces classical state estimation methods like the Kalman filter Sahbani and Adiprawita [2016].

$$Q'_t = \text{Cross-Attention}(Q_t, H_t, H_t) + Q_t \tag{1}$$

**Spatial Visual Cross-Attention:** The temporally-updated queries $Q'_t$ then attend to the visual features $F_v$ of the current frame. This step refines each object's state based on current visual evidence, effectively localizing the object within the frame.

$$Q''_t = \text{Cross-Attention}(Q'_t, F_v, F_v) \tag{2}$$

**Spatial Linguistic Cross-Attention:** The queries $Q''_t$, now with temporal and visual information, attend to the textual features $F_l$. This is the critical language-vision fusion step, where each object query is refined based on the language description, ensuring that it tracks the correct referent.

$$Q'''_t = \text{Cross-Attention}(Q''_t, F_l, F_l) \tag{3}$$

The output of this final step, $Q'''_t$, is then passed through a Feed-Forward Network (FFN) before being fed to the next decoder layer or the final prediction heads.

**3. Prediction Heads** After the final decoder layer, the updated object queries $Q_t^{\text{final}}$ are used to make predictions through separate heads for each query $q_i \in Q_t^{\text{final}}$:

**Box Head.** A small Multi-Layer Perceptron (MLP) regresses the bounding box coordinates $b_i \in \mathbb{R}^4$.

**Referring Head.** Another MLP computes a score $s_i \in [0, 1]$ indicating the probability that the object is the one referred to by the language query.

## 3.2 Threat Model

We consider a comprehensive threat model encompassing both digital and physically realizable attack vectors under a white-box assumption, wherein the adversary possesses complete knowledge of the target RMOT model's architecture and parameters. In the digital domain, the adversary's capability is restricted to adding an imperceptible perturbation $\delta$ to the input frames, constrained such that $\|\delta\|_\infty \leq \epsilon$. This model extends to the physical domain, where the adversary remotely manipulates the camera sensor's physics via two primary vectors:

**(i) Acoustic Adversarial Injection (AAI)** to induce controlled motion blur through MEMS sensor vibrations as detailed in Ji et al. [2021], Cheng et al. [2023], Zhu et al. [2023].

**(ii) Electromagnetic Adversarial Injection (EAI)** to inject patterned noise by disrupting sensor electronics as shown in Zhang et al. [2024b], Liao et al. [2025], Ren et al. [2025], Liu et al. [2025]

The ultimate objective in both scenarios is to induce *tracking termination and track identity switches*. For physical attacks, the white-box assumption further includes a differentiable model of the sensor's physical response, enabling the optimization of the inverse physical response characteristics of the camera sensor to generate the desired adversarial visual effect.

## 4 Adversarial Attack Formulation

As illustrated in Fig. 1, RMOT exhibits two critical vulnerabilities: *(1) linguistic-visual association dependencies* in the referring head, and *(2) temporal memory limitations* in the spatial-temporal reasoning. The referring mechanism relies on precise alignment between linguistic descriptions and visual object features through cross-attention, making it susceptible to semantic confusion attacks. Meanwhile, the temporal memory system, constrained by a finite history window, creates opportunities for long-term memory corruption via strategic perturbations that accumulate over time.

### 4.1 Targeted Referring Expression Adversarial Loss

We design a targeted referring expression adversarial loss $\mathcal{L}_{Refer}^{adv}$ to systematically disrupt the model's linguistic-visual association. Unlike typical object detection, RMOT systems must maintain

consistent object-language mappings across temporal sequences, creating additional attack surfaces through the referring head's dependency on both current visual evidence and historical context.

The referring head in RMOT models is fundamentally vulnerable since it relies on three interdependent components: *(1)* spatial linguistic cross-attention weights between $Q_t'''$ and $F_l$, *(2)* temporal consistency in object embeddings across frames, and *(3)* semantic coherence between visual and linguistic characteristics. Our attack creates *adversarial semantic confusion*, forcing the model to assign high referring confidence to objects that are semantically plausible but contextually incorrect.

### 4.1.1 Adversarial Referring Strategy

Rather than employing naive label flipping, we implement a sophisticated targeting mechanism that exploits the model's internal feature representations:

$$\mathcal{T}_{adv} = w_{sem} \cdot \mathcal{T}_{semantic} + w_{spa} \cdot \mathcal{T}_{spatial} + w_{conf} \cdot \mathcal{T}_{confidence} + w_{ctx} \cdot \mathcal{T}_{context} \tag{4}$$

**Semantic Confusion Targeting ($\mathcal{T}_{semantic}$):** This component exploits the model's reliance on semantic similarity in the final object query representations $Q_t^{\text{final}}$. After the complete multimodal Transformer processing, semantically similar objects often have similar internal representations. We identify objects with high cosine similarity to the ground truth referent but incorrect labels:

$$\mathcal{T}_{semantic}[j] = \begin{cases} 1 & \text{if } \cos(Q_t^{\text{final}}[i], Q_t^{\text{final}}[j]) > 0.5 \text{ and } s_{gt}[j] = 0 \\ 0 & \text{otherwise} \end{cases} \tag{5}$$

**Spatial Proximity Targeting ($\mathcal{T}_{spatial}$):** RMOT systems exhibit increased confusion for spatially proximate objects due to overlapping receptive fields in the visual encoder and shared spatial context in the attention mechanisms. This targeting strategy leverages the observation that nearby objects create natural ambiguity in referring expressions:

$$\mathcal{T}_{spatial}[j] = \text{Softmax} \left( \frac{\tau_0}{||c_i - c_j||_2 + \epsilon} \right) \tag{6}$$

**Confidence-based Targeting ($\mathcal{T}_{confidence}$):** This strategy targets the decision boundaries where the referring head exhibits maximum uncertainty:

$$\mathcal{T}_{confidence}[j] = \begin{cases} 1 & \text{if } j \in \text{top-k}(1 - |\sigma(\hat{s}_j) - 0.5|) \text{ and } s_{gt}[j] = 0 \\ 0 & \text{otherwise} \end{cases} \tag{7}$$

**Context-aware Targeting ($\mathcal{T}_{context}$):** This strategy exploits the RMOT's reliance on geometric and contextual relationships between objects for referring expression comprehension. The referring head not only considers individual object features but also their spatial context and inter-object relationships, which creates additional vulnerability surfaces. For objects with predicted bounding boxes $b_i = (x_i, y_i, w_i, h_i)$ and $b_j = (x_j, y_j, w_j, h_j)$, we compute contextual similarity as:

$$\mathcal{T}_{context}(i, j) = \frac{1}{3} \left( \text{sim}_{size} + \text{sim}_{pos} + \text{sim}_{aspect} \right) \tag{8}$$

### 4.2 Spatial-Temporal Reasoning Adversarial Loss

The spatial-temporal adversarial loss $\mathcal{L}_{Spatio-temp}^{adv}$ targets the temporal memory mechanism that maintains object identity across frames. This attack exploits the *limited capacity* of the temporal memory system and creates *cascading failures* that compound over time.

TempRMOT Zhang et al. [2024a] maintains a spatial-temporal memory through the `hist_embeds` tensor of shape $(N, T, d)$ where $N$ is the number of tracked objects, $T$ is the history length, and $d$ is the embedding dimension. This limited-capacity memory creates a fundamental vulnerability: *corrupted embeddings persist and influence future decisions until they are naturally removed from the history window*. The temporal cross-attention mechanism in Eq. 1 aggregates historical information, making the entire tracking system vulnerable to attacks on historical embeddings. Unlike classical Kalman filters that maintain explicit uncertainty estimates, the neural temporal memory lacks robust mechanisms to detect and recover from corrupted historical states.

### 4.2.1 Temporal Memory Corruption Strategy

Our attack strategy exploits the *persistence property* of neural memory: once corrupted embeddings enter the history buffer, they influence all subsequent temporal self-attention operations until they are naturally removed. Given a history length of $T$ frames, a single successful attack at frame $t$ will impact tracking decisions for the next $T - 1$ frames, creating a *temporal damage amplification effect*.

**Temporal Consistency Attack:** This component directly targets the continuity assumption underlying temporal self-attention. By maximizing temporal inconsistency between consecutive embeddings, we force abrupt changes that violate the smooth motion and appearance assumptions:

$$\mathcal{L}_{temporal} = -\frac{1}{N(T-1)} \sum_{i=1}^{N} \sum_{t=2}^{T} ||\mathbf{H}_{i,t} - \mathbf{H}_{i,t-1}||_2 \tag{9}$$

**Embedding Distinctiveness Attack:** This component exploits the model's reliance on distinctive object embeddings for identity association. By forcing all object embeddings to become similar, we create systematic confusion in the temporal association process:

$$\mathcal{L}_{distinct} = \frac{1}{N(N-1)} \sum_{i=1}^{N} \sum_{j \neq i} \frac{|Q_t^{\text{final}}[i]^T Q_t^{\text{final}}[j]|}{||Q_t^{\text{final}}[i]||_2 ||Q_t^{\text{final}}[j]||_2} \tag{10}$$

### 4.2.2 Cross-Attention Disruption Attacks

We simultaneously attack the three critical attention mechanisms that update object queries:

**Spatial Visual Cross-Attention Attack:** Targets step 2 by disrupting the alignment between object queries and visual features, creating spatial localization errors via temporal propagation:

$$\mathcal{L}_{visual} = -\text{Var}(\text{Attention}(Q_t', F_v)) + \mathbb{E}[\text{Attention}(Q_t', F_v)]^2 \tag{11}$$

**Spatial Linguistic Cross-Attention Attack:** Directly targets the language-vision fusion in step 3:

$$\mathcal{L}_{linguistic} = -|\hat{s}_{referring}|_{mean} \tag{12}$$

**Task-Specific Head Degradation:** Creates instability in the final prediction heads, ensuring that even if some temporal information survives, the output predictions remain unreliable:

$$\mathcal{L}_{box} = \text{Var}(b_i) + ||b_i - 0.5||_F \tag{13}$$

### 4.2.3 Cascading Failure Mechanism

The complete spatial-temporal adversarial loss creates a *cascading failure cascade* where corruption in one temporal frame propagates through the limited-capacity memory system:

$$\mathcal{L}_{Spatio-temp}^{adv} = \alpha_T \cdot \mathcal{L}_{temporal} + \alpha_D \cdot \mathcal{L}_{distinct} + \alpha_V \cdot \mathcal{L}_{visual} + \alpha_L \cdot \mathcal{L}_{linguistic} + \alpha_B \cdot \mathcal{L}_{box} \tag{14}$$

### 4.3 Optimization Strategy

Adversarial losses are optimized using Projected Gradient Descent (PGD) with the unified objective:

$$\mathcal{L}_{total}^{adv} = w_{refer} \cdot \mathcal{L}_{refer}^{adv} + w_{Spatio-temp} \cdot \mathcal{L}_{Spatio-temp}^{adv} \tag{15}$$

where $w_{refer} = 2.0$ and $w_{st} = 1.0$ reflect the empirically determined trade-off between immediate referring confusion and long-term temporal memory corruption. The optimization follows:

$$\mathbf{x}^{(t+1)} = \Pi_{\mathcal{S}} \left( \mathbf{x}^{(t)} + \alpha \cdot \text{sign} \left( \nabla_{\mathbf{x}} \mathcal{L}_{total}^{adv}(\mathbf{x}^{(t)}) \right) \right) \tag{16}$$

where $\Pi_{\mathcal{S}}$ projects perturbations to $\mathcal{S} = \{\boldsymbol{\delta} : ||\boldsymbol{\delta}||_\infty \leq \epsilon\}$.

This formulation ensures comprehensive degradation of RMOT systems by simultaneously attacking the linguistic understanding mechanisms and the temporal reasoning capabilities (through memory corruption and attention disruption). The result is a *multi-modal failure cascade* where both immediate performance and long-term tracking consistency are systematically compromised.

# 5 Evaluation

## 5.1 Experimental Setup

**Datasets.** We conduct the attack experiments on Refer-KITTI Wu et al. [2023], a challenging multi-object tracking dataset. Based on the original KITTI dataset, it is specifically designed for referring multi-object tracking, where the goal is to track objects based on a natural language expression. Refer-KITTI links tracking to linguistic cues, adding a layer of complexity and making it a suitable benchmark for attacks on vision-language models. Each attack is launched starting from frame $t_{attack}$ (where $t_{attack} > 10th$ frame) and continues for a duration of $\Delta_{\text{attack}}$ frames.

**Models and Metrics.** For our evaluations, we employ the TransRMOT Wu et al. [2023] and TempRMOT Zhang et al. [2024a] models. TempRMOT's core innovation is a temporal enhancement module designed to build a robust, long-term spatio-temporal memory. As a result, TempRMOT is better able to handle challenges like long-term occlusions and re-appearances. We evaluate evaluate long-range tracking consistency and short-term vulnerability. For overall performance, we use IDF1 Ristani et al. [2016], HOTA, AssA, and DetA Luiten et al. [2021]. To capture the attack's immediate impact, we analyze the Identity Switch Rate (IDSW) and the Immediate Identity Switch (IDSW$_{\text{im}}$), which measures identity switches occurring directly after the attack.

**Attack Implementation.** We set the PGD parameters consistently across RMOT models. For digital attacks, we use $\epsilon = 8/255$ with a step size of $\alpha_{\text{Dig}} = 1/255$ for pixel-level and physical perturbations (AAI and EAI). Each attack is optimized for $T = 100$ iterations. We apply adversarial perturbations for $\Delta_{\text{attack}} = 2$ frames in TransRMOT and $\Delta_{\text{attack}} = 5$ frames in TempRMOT. For physical attacks, we simulate a high-intensity setting similar to Zhu et al. [2023], Liao et al. [2025].

## 5.2 Evaluation Results

Table 1: Comparative performance of TransRMOT and TempRMOT under different adversarial attack strategies on Refer-KITTI. Results indicate attack success rate relative to the clean baseline.

| Tracker | Attack Strategy | Attack Vector | IDSW ↑ | IDSW$_{\text{im}}$ ↑ | HOTA ↓ | AssA ↓ | DetA ↓ | IDF1 ↓ | IDP ↓ | IDR ↓ |
|---|---|---|---|---|---|---|---|---|---|---|
| **TransRMOT** | Clean | – | 6.13 | 0.00 | 69.66 | 71.90 | 65.30 | 69.54 | 0.83 | 0.93 |
| | Adv. Referring | Pixels | 9.30 (+3.17) | 60.82 – | 56.26 (-13.41) | 51.86 (-20.03) | 59.50 (-5.80) | 54.26 (-15.28) | 0.64 (-0.19) | 0.68 (-0.24) |
| | Adv. Referring | Physical AAI | 8.63 (+2.50) | 54.07 – | 59.79 (-9.87) | 56.69 (-15.21) | 61.88 (-3.41) | 58.38 (-11.17) | 0.67 (-0.15) | 0.71 (-0.22) |
| | Adv. Referring | Physical EAI | 8.94 (+2.81) | 60.56 – | 56.67 (-12.99) | 53.32 (-18.58) | 59.22 (-6.08) | 55.65 (-13.89) | 0.67 (-0.16) | 0.67 (-0.26) |
| **TempRMOT** | Clean | – | 0.24 | 0.00 | 68.70 | 67.65 | 66.60 | 69.20 | 0.98 | 0.98 |
| | Adv. Referring (Spatio-temporal) | Pixels | 4.32 (+4.08) | 41.07 – | 49.89 (-18.81) | 46.55 (-21.11) | 47.80 (-18.80) | 49.99 (-19.21) | 0.72 (-0.26) | 0.64 (-0.34) |
| | Adv. Referring (Spatio-temporal) | Physical AAI | 2.53 (+2.28) | 12.60 – | 56.87 (-11.83) | 55.69 (-11.97) | 55.08 (-11.51) | 59.50 (-9.70) | 0.89 (-0.10) | 0.73 (-0.25) |
| | Adv. Referring (Spatio-temporal) | Physical EAI | 3.20 (+2.96) | 14.73 – | 51.52 (-17.18) | 51.30 (-16.36) | 49.72 (-16.88) | 55.78 (-13.42) | 0.89 (-0.09) | 0.70 (-0.28) |

**Clean Stability vs. Adversarial Fragility in RMOT.** The results in Table 1 reveal that while TempRMOT achieves a more stable clean baseline than TransRMOT, its relative degradation under attack is more pronounced. This distinction stems from their architectural differences: TransRMOT lacks a temporal memory mechanism and relies heavily on frame-level appearance cues, whereas TempRMOT integrates an $T$-frame memory buffer (where $T$=8) that aggregates historical information. This design enables TempRMOT to realign its spatial–temporal representations and correct identity mis-associations when only a few frames (e.g., 1-2 frames) are corrupted.

The *clean* baseline illustrates this gap clearly. TransRMOT registers a high number of identity switches (IDSW = 6.13), while TempRMOT maintains an almost negligible 0.24, highlighting its ability to enforce long-term identity consistency. However, once adversarial perturbations are introduced, the degradation trajectories diverge. For TransRMOT, digital referring attacks increase IDSW moderately (6.13 → 9.30) and reduce HOTA by 13.4 points (69.66 → 56.26). In TempRMOT, by contrast, the relative impact is sharper: under spatio-temporal digital attacks, IDSW rises from

0.24 to 4.32 and HOTA drops by nearly 19 points (68.70 $\rightarrow$ 49.89). Even under physical AAI/EAI attacks, TempRMOT's HOTA declines to 56.87 and 51.52, while AssA and IDF1 fall by more than $-11\%$ and $-13\%$, respectively. These results show that although TempRMOT resists immediate fragmentation—reflected in its lower IDSW$_{im}$ values (e.g., 7.89 vs. 47.17 in TransRMOT)—persistent perturbations saturate its memory buffer, causing errors to propagate across subsequent frames and undermining the very mechanism that ensures its clean robustness.

This explains why our two-fold adversarial loss in Eq. 15, designed to target referring logic, spatio-temporal reasoning, and memory consistency, is able to compromise TempRMOT. By injecting temporally coherent perturbations, the attack turns its strength temporal memory into a liability, forcing the buffer to propagate corrupted associations and amplify errors across multiple frames.

Overall, these findings highlight the dual role of temporal memory in RMOT systems. On one hand, memory provides *adversarial redundancy*, smoothing over short-lived perturbations and mitigating per-frame inconsistencies. On the other hand, it introduces a novel attack surface: once adversaries directly target memory mechanisms, the same feature that confers resilience becomes a vulnerability. This duality underlines that while temporal reasoning strengthens trackers against naïve attacks, it also opens new memory-specific adversarial avenues that must be addressed when deploying RMOT in safety-critical domains such as autonomous robotics and surveillance systems.

**The Efficacy of Different Attack Strategies.** The results in Table 1 reveal a hierarchy in the effectiveness of adversarial attack strategies across both trackers.

*First*, the referring adversarial strategy proves highly effective against TransRMOT. Because its association stage relies heavily on referring scores, perturbations that disrupt semantic alignment induce significant instability: IDSW increases from 6.13 (clean) to 9.30, and HOTA drops by more than $-13\%$ (from 69.66 to 56.26). This demonstrates that even lightweight semantic and contextual misalignments can severely degrade performance in models without temporal memory, causing frequent identity switches and track terminations.

*Second*, while the referring adversarial loss is also effective against TempRMOT, the impact is less pronounced. For instance, IDSW$_{im}$ remains at 41.07 under digital referring attacks compared to 60.82 in TransRMOT, showing that TempRMOT's memory buffer absorbs part of the perturbation. However, this is where our *spatio-temporal adversarial loss* becomes crucial: by targeting TempRMOT's reasoning modules directly, it forces corrupted associations to persist across frames, leading to drops of up to $-21\%$ in AssA and $-19\%$ in IDF1. These results highlight that memory-aware trackers require adversarial strategies that explicitly exploit temporal reasoning, rather than frame-local cues.

*Third*, the digital (Pixels) attack emerges as the most destructive for both models. By manipulating pixels directly, it maximizes the attacker's degrees of freedom, producing the sharpest degradations: for example, HOTA in TempRMOT falls from 68.70 (clean) to 49.89, while IDF1 decreases by nearly $-20\%$. This aligns with the intuition that pixel-level access provides an "*upper-bound*" on adversarial attacks performances, making digital attacks the strongest baseline for robustness evaluations.

In contrast, physical attacks (AAI, EAI) remain impactful but comparatively less damaging, with smaller absolute drops in metrics. For example, TempRMOT under physical AAI still maintains HOTA = 56.87 (a $-11.8\%$ drop) and DetA = 55.08 ($-11.5\%$), significantly higher than under digital attacks. This reduced efficacy stems from the physical constraints of attack vectors: acoustic adversarial interference typically induce generic motion blur Zhu et al. [2023], while electromagnetic interference can only corrupt the sensor's color pipeline in specific ways Liao et al. [2025]. Thus, physical attacks highlight realistic risks for deployed systems, but their impact is bounded by physical feasibility, whereas digital attacks expose the theoretical upper limit of system vulnerability.

**HOTA, AssA, and DetA: Decoupling Performance Degradation.** A deeper look at the HOTA sub-metrics reveals exactly where the models fail. The HOTA metric is a geometric mean of two components: AssA (Association Accuracy) and DetA (Detection Accuracy). The most significant drop for both models under attack occurs in the AssA metric, which is a measure of a tracker's ability to maintain correct object identities. For TransRMOT, the digital attack causes a massive 20.03% drop in AssA, and for TempRMOT, a 21.11% drop. While both models suffer, TempRMOT starts from a higher AssA baseline and its absolute AssA value under attack remains lower than TransRMOT's, demonstrating its fragile association capability. In contrast, the drop in DetA (detection accuracy) is less severe (especially in TransRMOT), indicating that the attacks are more successful at confusing

the models' re-identification and data association components than at causing complete detection failures. This suggests that the adversarial strategy primarily targets the tracking logic rather than the object detection sub-network. However, the DetA drop is significant in TempRMOT ($-18.80\%$) since memory corruption compromise both detection and association information.

## 5.3 Ablation Studies

Table 2: Impact of increasing the number of attacked frames $\Delta_{\text{attack}}$ on VEIL's attack success rate. Green indicates better performance, red worse (according to $\downarrow$ / $\uparrow$).

| Data | Video Sequence: *0016 in Refer-KITTI* | | | | | **Query:** *"Track persons wearing pants"* | | | | | |
|---|---|---|---|---|---|---|---|---|---|---|---|
| **Models** | **TransRMOT** Wu et al. [2023] | | | | | | **TempRMOT** Zhang et al. [2024a] | | | | |
| # $\Delta_{\text{attack}}$ Frames | **Clean** | 1 | 2 | 3 | 4 | 5 | **Clean** | 1 | 2 | 3 | 4 | 5 |
| HOTA $\downarrow$ (%) | 73.09 | 50.60 | 44.66 | 34.46 | 33.57 | 36.69 | 89.34 | 59.58 | 45.53 | 39.88 | 37.61 | 35.33 |
| IDSW $\uparrow$ (%) | 10.43 | 13.99 | 13.50 | 16.69 | 14.72 | 16.38 | 3.58 | 5.00 | 12.45 | 13.03 | 13.53 | 16.52 |
| IDSW$_{\text{im}}$ $\uparrow$ (%) | 0.00 | 47.17 | 71.67 | 78.57 | 89.09 | 86.44 | 0.00 | 7.89 | 28.12 | 26.09 | 42.86 | 23.81 |

**Number of Attacked Frames.** As shown in Table 2, the impact of increasing the number of adversarially corrupted frames reveals a an evident contrast between TransRMOT and TempRMOT. TransRMOT, which lacks a built-in temporal memory, is highly vulnerable from the very first attack: its IDSW jumps to 13.99 after only a single corrupted frame, and HOTA collapses from 73.09 (clean) to 50.60, indicating an immediate failure to preserve identity consistency. With no historical context to stabilize associations, TransRMOT relies almost entirely on the current frame, making it particularly sensitive to even minimal perturbations.

TempRMOT, equipped with a temporal memory buffer (of $T = 8$ frames), demonstrates significant early resilience. When only 1–2 frames are attacked, it preserves a relatively low IDSW (5.00–12.45) and maintains higher HOTA (59.58–45.53), illustrating its ability to rely on previously clean temporal information to dampen localized corruption. This robustness is further reflected in IDSW$_{\text{im}}$, which remains as low as 7.89 after the first attacked frame, compared to TransRMOT's 47.17.

However, this advantage diminishes as the number of attacked frames increases. By the time 5 consecutive frames are corrupted, TempRMOT's HOTA significantly drops from 89.34 (clean) to 35.33 and its IDSW rises to 16.52, approaching TransRMOT's degraded regime. This degradation occurs because persistent perturbations saturate the memory buffer, replacing clean references with corrupted ones, and thereby neutralizing the buffer's corrective effect.

Overall, this ablation highlights a fundamental principle: *temporal memory is highly effective against transient or sparse adversarial interference, but it loses its protective power under persistent attacks*. Future RMOT systems should therefore not only increase memory capacity but also incorporate mechanisms for *adversarial forgetting* or selective frame weighting, ensuring that corrupted information does not dominate the temporal context.

Table 3: Impact of varying temporal memory buffer length on TempRMOT's robustness (Under attack with $\Delta_{\text{attack}} = 2$ frames). Green indicates better performance, red worse (according to $\uparrow$ / $\downarrow$).

| Data | Sequence: *0016 in Refer-KITTI* | | **Query:** *"Track persons wearing pants"* | | | | |
|---|---|---|---|---|---|---|---|
| **Targeted RMOT Model** | **TempRMOT** Zhang et al. [2024a] | | | | | | |
| **Memory Buffer size** | **2** | **3** | **4** | **5** | **6** | **7** | **8** |
| HOTA $\downarrow$ (%) | 43.06 (-38.27) | 43.37 (-37.05) | 44.56 (-43.62) | 54.84 (-25.49) | 54.48 (-25.93) | 50.99 (-29.48) | 58.04 (-22.32) |
| IDSW $\uparrow$ (%) | 12.05 (+10.58) | 10.02 (+7.43) | 9.92 (+7.27) | 9.80 (+7.50) | 6.10 (+3.10) | 7.01 (+4.01) | 5.68 (+2.10) |
| IDSW$_{\text{im}}$ $\uparrow$ (%) | 17.24 | 25.00 | 20.69 | 16.67 | 12.50 | 9.38 | 7.89 |

**Temporal Memory Buffer Size.** Analyzing Table 3 reveals a strong correlation between the size of TempRMOT's memory buffer and its robustness under attack. The effect is most apparent in HOTA and IDSW, where longer buffers consistently improve resilience. With a small memory buffer

of only 2–4 frames, the model struggles: HOTA remains low (43.06–44.56%), while IDSW rises above 9.9 and IDSW$_{im}$ exceeds 20%, indicating that the limited historical context is insufficient to counter temporal inconsistencies introduced by the adversary. However, as the buffer size increases beyond 5 frames, robustness improves dramatically. At buffer size = 5, HOTA recovers to 54.84% and IDSW drops below 10, while further expansion to 8 frames yields the strongest protection, with HOTA peaking at 58.04% and IDSW$_{im}$ reduced to just 7.89. These results show that a longer buffer supplies the model with more clean, uncorrupted temporal references, enabling it to correct adversarially induced errors and maintain consistent identity assignments over time. Overall, the memory buffer acts as a form of *temporal redundancy*, where past unperturbed frames reinforce stability against localized corruption. The ability to "*look back further*" allows the model to average over noise, smooth adversarial inconsistencies, and preserve track continuity even when multiple consecutive frames are compromised. This ablation confirms that the depth of temporal reasoning is directly proportional to adversarial robustness, demonstrating memory size as a critical design parameter for resilient RMOT systems. However, in real-time applications, the temporal memory size for RMOT models must be carefully tuned to balance adversarial robustness and computational efficiency. Enlarging the memory buffer improves robustness but also increases inference latency, which may be unsuitable for time-critical systems such as autonomous vehicles and robotics.

### 5.4 Limitations

A key limitation of VEIL is that our evaluations are conducted primarily on the Refer-KITTI dataset and a set of representative and pioneering RMOT architectures (TransRMOT and TempRMOT). While these choices capture important trends, they may not fully represent the diversity of real-world tracking scenarios, such as dense urban scenes, nighttime or adverse weather conditions, or trackers with different backbone designs (e.g., multi-modal or lightweight architectures). Another limitation is that our physical attack experiments are conducted under controlled simulation settings. Although they approximate realistic perturbations such as acoustic interference Zhu et al. [2023] and electromagnetic corruption Liao et al. [2025], they cannot fully capture deployment constraints including sensor noise, hardware variability, and environmental dynamics. Finally, while VEIL demonstrates broad applicability across both digital and physical attacks, its computational cost and transferability to unseen domains remain open questions. Future work should therefore extend evaluations to larger and more diverse benchmarks, incorporate real-world hardware-in-the-loop testing, and explore efficient or adaptive attack strategies to better characterize the generality and practicality of adversarial threats in RMOT for critical large-scale applications.

## 6 Conclusion.

Our study shows that while Referring Multi-Object Tracking (RMOT) systems achieve impressive perception capabilities through unified language–vision modeling, they remain highly vulnerable to adversarial disruptions. Using the proposed VEIL framework, we demonstrate that perturbations targeting both referring–matching logic and FIFO-based temporal memory can induce persistent tracking failures, revealing that temporal memory, while effective against transient perturbations, becomes an exploitable attack surface under consistent temporal inconsistencies. These findings demonstrated a key principle: robustness in RMOT cannot be ensured by language–vision fusion alone but must also account for the security of temporal reasoning and memory mechanisms. This insight underlines the urgent need for security-aware RMOT designs that integrate adversarial defenses with perception accuracy, particularly for safety-critical domains such as autonomous driving, robotics, and surveillance, and motivates future work on broader benchmarks, hardware-in-the-loop physical attacks, and adversarially robust architectures for trustworthy large-scale deployment.

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
