# OpenReview forum: "Adversarial Attacks against Context-dependent Visual Association in Referring Multi-Object Tracking Systems"
_NeurIPS.cc/2025/Workshop/Reliable_ML — NeurIPS 2025 - Reliable ML Workshop_

### Official Review · Reviewer_AkEH · 2025-09-24

**Rating:** 6
**Confidence:** 4

**Review:**

The paper introduces VEIL, an adversarial attack framework that targets Referring Multi-Object Tracking (RMOT) systems, which unify language-vision fusion, spatial reasoning, and temporal memory in multimodal Transformers. VEIL exploits two vulnerabilities: (1) linguistic-visual association in the referring head, and (2) FIFO-based temporal memory in spatio-temporal reasoning. The attack combines targeted semantic confusion with temporal inconsistency to induce identity switches, track terminations, and cascading memory corruption. Experiments on Refer-KITTI with TransRMOT and TempRMOT show that VEIL significantly degrades tracking performance: e.g., TransRMOT suffers up to 20% drop in association accuracy and large increases in ID switches, while TempRMOT—though more resilient due to memory buffers, still experiences severe performance degradation under digital and physical attacks (acoustic and electromagnetic perturbations). Ablations confirm that longer temporal memory improves robustness, but cannot fully prevent failures.

Strengths:
First work according to my knowledge to systematically study adversarial vulnerabilities of E2E Transformer-based RMOT systems.
Provides a detailed attack formulation combining linguistic confusion and temporal reasoning corruption.

Weaknesses:
"Linguistic" may not be the most indicative word for describing the method. "Context-dependent" may be better.
Limited experiments on only one dataset.